# On the Reality of Signaling in Auctions

Aviad Levi and Shani Alkoby *

Department of Industrial Engineering and Management, Faculty of Engineering, Ariel University,
Ariel 4070000, Israel
* Correspondence: shania@ariel.ac.il

**Abstract:** Over the last two decades, auctions have become an integral part of e-commerce and a promising field for applying artificial intelligence technologies. The use of signals has been studied extensively in the existing auction literature. Specifically, it has been shown that when an external strategic entity (such as an information broker) is present, it can be beneficial to use signaling as a preliminary step before offering to sell information. However, these results apply only in cases where all auction participants are completely rational agents. However, in many real-life scenarios some of the participants are humans, and hence are easily affected by external factors, i.e., their rationality is bounded. In this paper, we offer a thorough investigation of a case in which the prospective information buyer is a human auctioneer. Using a set of MTurk-based experiments with people, we tested 10,000 independent auctions with diverse characteristics, and were able to identify a varied set of practical insights regarding human behavior. Real-life strategic information brokers could potentially use these insights to achieve a better understanding of how humans operate, paving the way for optimizing the benefit obtainable from the information they own.

**Keywords:** information in auctions; signals; value of information; human–computer interaction; expert; information broker; bounded rationality; game theory



## 1. Introduction

Auctions are one of the most popular and effective trading methods currently in use. Their popularity stems from the fact that in most cases, they result in efficient allocation, are an effective tool for allocating scarce or perishable goods, and facilitate effective extraction of the bidders' evaluations [1]. Thus, in the last two decades, auctions have been widely discussed in the literature on both economics [2,3] and computer science [4,5]. Auctions are also used for many practical applications such as spectrum auctions, stock exchanges, e-commerce websites (e.g., eBay), online advertising, procurement auctions for governments, electricity markets, wireless sensor networks, art, and more.

Many times, auctions include some uncertainty associated with the auctioned item. Most commonly, the uncertainty relates to the bidders' knowledge regarding the value of the auctioned item, meaning that at the time of the auction, bidders have only an estimate of the item's value. As a result, information revelation has come to play a central role in the auction environment. One recently popular line of research takes this type of uncertainty a step forward, and assumes the existence of an additional entity in the auction, an "information broker," who holds information that can disambiguate the uncertainty concerning the auctioned item. This research line usually assumes that the auctioneer is also unfamiliar with the exact value of the auctioned item, i.e., the information broker is the only one with this information. Being an independent entity, the information broker often (but not always) is interested in maximizing their own personal revenue from the information, and thus offers to sell it for the highest price possible. It has already been shown that in many cases, the information broker finds it beneficial to use signaling to partially reveal some of the information, for free, before offering to sell the information. By

doing this, the information broker encourages buyers to purchase the information in more cases and for a higher price [6].

The problem of an information broker, who uses signals in an attempt to provide potential information buyers with a more accurate characterization of the item's value, was discussed in a recent paper [7]. The model studied there considered all of the auction participants to be completely rational agents. In many real-world auctions, however, we expect to find people, i.e., agents with bounded rationality, in the role of the auctioneer. We thus, hypothesize that—as opposed to the case of a completely rational auctioneer, in which one can eventually calculate the resulting equilibrium and anticipate the agents' future behavior—when it comes to people, the auctioneer's behavior becomes unpredictable. Hence, the results presented in previous work may not be relevant, and a self-interested information broker might not be able to use the same strategies and techniques used in the case of a completely rational auctioneer. To reach a realistic characterization of human auctioneers' behavior, we provide a thorough experimental evaluation of human behavior and performance in this somewhat contradictory, dual role of item seller and prospective information buyer.

Our contribution is twofold. First, we uncover another tier in the gap between theory and reality, and reveal new bounded rational behaviors supported by experimental evidence. In doing so, we provide a new rationale showing that a self-interested information broker in an auction should sometimes deviate to more human-suitable behavior (which may contradict the suggested rational behavior). Secondly, from our empirical data, we distill a versatile set of practical, useful insights that could be helpful to a real-life information broker. By translating those insights into strategic behavior, a self-interested broker might be able to optimize their expected revenue from the information they own.

The paper is organized as follows. Section 2 provides a comprehensive review of the relevant literature. Next, we focus on setting up the model in Section 3. Section 4 introduces some preliminary issues and discusses the changes and adjustments required to deal with a human information buyer. Thereafter, Section 5 presents an empirical study meant to test human behavior. Finally, Section 6 presents our conclusions and proposals for future work.

## 2. Related Work

Strategic signaling and optimal free disclosure of information in auctions have been studied widely in literature on both economics [8–11] and artificial intelligence (AI) [12–14]. The work of Fishman [8] discusses the problem of a takeover bidding process with multiple bidders. In this paper, bidders can choose to be exposed to the target's private information for a cost. Based on the information disclosed to them, the bidder decides whether to make a takeover bid. The work of Daniel and Hirshleifer [9] models sequential bidding in an English auction when submitting or revising a bid is costly. In their work, bidders can "signal" their high valuation by jumping over previous bids. The work of Dodonova [10] extends the work of Fishman and shows that by adding a common value component, the resulting social surplus is higher. Finally, the work of Dodonova and Khoroshilov [11] shows that preemptive jump bidding when the other bidder owns a fraction of the target firm leads to a higher social surplus and improves the expected profit of both bidders. As opposed to our paper, in which the use of signals is being initiated by the expert (an external entity), in all these studies, the bidders themselves are the initiators, choosing whether to be exposed to additional information.

Milgrom and Weber [15] showed that a seller's revenue increases as more information, positive or negative, is disclosed to the bidders (the "linkage principle"). Conversely, others have shown that for some settings, full disclosure of the information might not be ideal for a self-interested auctioneer [16–18]. One essential question in the context of auctions concerns the optimal auction design. Myerson [19] and Riley and Samuelson [20] provided evidence for the equivalence of different auction types if no signals are used (e.g., the first-price and second-price equivalence). This is not necessarily the case when signals are used. Giovannoni and Makris [21] showed that different auction formats combined with varying disclosure policies yield diverse auction revenue.

As opposed to our research, in which signals are being used by the information broker, many works have considered problems in which the item seller (i.e., the auctioneer) is the one using signals [22–24]. Board [17] showed that in the case of a second-price, sealed-bid auction that includes only two bidders, the auctioneer should not reveal any information. Bergemann and Pesendorfer [16] considered a model where signals are being used by the seller to allow the bidders to learn their own valuation of the auctioned item. Li and Shi [18] analyzed a two-period screening model where the auctioneer could disclose information correlated with the buyer's private type. Ganuza and Penalva [25] examined the auctioneer's incentive to reveal private information, and Arefeva and Meng [26] considered a model in which the auctioneer has information that affects the bidders' independent, private valuations additively. Additionally, by allowing the bidders themselves to use signals (e.g., by placing different bids during the auction) one can dramatically influence the participants' behavior both during and after the auction [27–29].

Optimal free information disclosure (specifically in second-price auctions) is used in various fields, including real estate markets [30], digital advertising [31], distribution platforms [32], home energy efficiency [33], and supply-chain management [34,35]. Due to it being a useful tool in practice, and considering the fact that humans are its target population, there is a growing need to have a better understanding of human behavior in such environments and whether the existing theory still holds.

The "value of information" (usually denoted "*VoI*") is a mathematical formulation used to quantify the potential benefit of additional information in the face of uncertainty. *VoI* can be formally defined using two alternative perspectives: (1) assessing a potential user's willingness to pay for obtaining access to the knowledge [34,35]; or (2) the increase in expected benefit that arises from acting according to the best choice with the presence of the information compared to the expected benefit when no information is available [34,36]. The concept of *VoI* has become highly relevant now that information sharing is a daily activity; indeed, *VoI* analysis is used in varied fields, including health-economic evaluations [37], supply-chain decisions [38], online knowledge payment platforms [39], earth sciences [40], and auction research [41,42]. *VoI* can be measured both in cases with full disclosure, i.e., the information provided eliminates the uncertainty entirely [43], and in those with partial disclosure in which the information provided eliminates only part of the uncertainty [44]. While the subject of information disclosure is broadly discussed in the existing literature, most economic models assume full disclosure and complete rationality [26,45,46]. In many real-life situations, however, agents (i.e., people) are faced with partial information and make decisions under bounded rationality. The premise of bounded rationality contradicts the feasibility of the existing models of perfect rationality aiming to solve multiagent problems [47].

Many prior studies have examined problems of bounded rationality [48,49], and specifically the combination of information disclosure in auctions and bounded rationality. A majority of those works, however, have concentrated on the overbidding phenomenon, e.g., bidders' tendency to overbid when they are facing the possibility of receiving (free or costly) signals concerning their opponents [50,51]. Our study has some resemblance to the experimental work of Andreoni et al. [52], who studied how bidders' knowledge of their rivals' types affects their behavior in the auction. However, their experimental design is meant to examine how bidders, who are familiar with their own valuation of the auctioned item, respond to information regarding their rivals' valuation. In our work, the uncertainty is more complex because it includes a lack of awareness regarding the actual world state, which influences bidders' knowledge regarding both their own and their rivals' valuation of the auctioned item. Moreover, our work does not resolve the uncertainty regarding rival bidders' valuations.

Finally, this study was built on the basis of several past studies [7,53]. The work of Alkoby et al. [7] also considered a strategic use of signals by a self-interested expert in the context of auctions. In their work, however, all participants, including the auctioneer, are assumed to be completely rational agents. Hence, they offer a three-party equilibrium anal-

ysis, in which the expert uses signals based on a pre-committed scheme before presenting their price for revealing the exact information. As a result, no experimental investigation was needed. In the work of Alkoby and Sarne [53], a series of MTurk-based experiments were conducted to test the benefit of free public information disclosure. Still, their work considered a much simpler decision-making problem in which the information buyer was isolated, rather than part of a complex multiagent system, like an auction. To the best of our knowledge, there have been no prior empirical investigations focusing on the benefit of the use of signaling by a self-interested expert facing an auctioneer with bounded rationality.

## 3. Model

We consider a standard second-price, sealed-bid auction with an auctioneer, $a$, a finite set of $n$ bidders, $B = \{b_1, b_2, b_3, \ldots, b_n\}$, and one auctioned item, $I$. The value of the auctioned item is a priori unknown, and is affected by the world state set by nature. The set of possible world states, denoted by $WS$, is finite, i.e., $WS = \{ws_1, ws_2, \ldots, ws_m\}$, and each world state's probability of being the actual one is defined as $p(ws_i)$ (where $\sum_{ws_i \in WS} ws_i = 1$). Each bidder is assumed to be characterized by a type, $t$. The set of possible types, denoted by $T$, is finite, i.e., $T = \{t_1, t_2, \ldots, t_k\}$. All bidder types are assumed to be independent and identically distributed, such that the a priori probability of any of the bidders being of type $t_i$ is given by $p(t_i)$ (where $\sum_{t_i \in T} t_i = 1$). A bidder's type defines their valuation of the auctioned item for any possible world state. The bidders' valuations of the possible world states can be compactly represented in the form of a matrix, $V$, of size $k \times m$, where $V[j][i]$ contains type $j$'s valuation of $ws_i$. Each bidder is assumed to know their own type, but not the type of the other bidders taking part in the auction. Similarly, the auctioneer is assumed to be unfamiliar with the specific type of each bidder participating in the auction, and is only aware of the types' probabilities.

In addition to the auctioneer and the bidders, we assume the existence of an external, independent entity who has access to the information regarding the actual world state. This entity can be referred to as an "expert" and will be denoted by $E$. The expert does not a priori own the information regarding the actual world state, but has the capabilities needed to mine it. An example of such an expert would be an appraiser who can provide the exact manufacturing date of an antique that a seller is interested in selling. Knowledge of the actual world state gives the expert an opportunity to influence the course of the auction by causing bidders to change their bids. Not knowing the actual world state, bidders' bids will be equal to the expectation of their valuations for all the different possible world states. However, a bidder aware of the actual world state will bid an amount equal to their valuation of the item for the specific world state. The expert offers to sell the information to the auctioneer in exchange for a cost, $c$.

In our model, only the auctioneer has the option of purchasing the information; bidders by themselves are not allowed or are unable to acquire the information. If the auctioneer decided to purchase the information, however, they are obligated to disclose it to the bidders. This assumption has a foundation in reality, as regulations governing many real-life situations mandate full disclosure of information. Hence, bidders are exposed to the actual world state only because the auctioneer chose to purchase the information.

We assume all participants are aware of the number of bidders taking part in the auction ($n$), the possible world states and their probabilities ($WS$ and $\forall ws_i \ p(ws_i)$), the possible bidder types and their probabilities ($T$ and $\forall t_j \ p(t_j)$), and the valuation of each bidder type for every possible world state ($V$). The auctioneer, bidders, and expert are assumed to be self-interested, i.e., they all attempt to maximize their expected profit from the auction. The auctioneer's expected profit is defined as the profit from the auction (i.e., the expected second-best bid) minus the payment $c$ if they choose to purchase the information. A bidder's profit is their valuation of the item minus their payment to the auctioneer if they win the auction and zero otherwise. The expert's expected profit is determined solely on the basis of whether or not they sold the information (and the price $c$ set by them) and the cost of obtaining this information. For simplicity, we will assume

that mining this information has no cost for the expert. In those cases where mining the information has a cost (e.g., it is time-consuming), the monetary value of the cost can be included in the relevant calculations. Finally, to increase their expected profit, the expert can act strategically in an attempt to maximize the chances that the information will be purchased. For this purpose, the expert can use signals to freely disclose information and publicly eliminate some of the possible world states. Practically, assuming the actual world state is $ws_i$, the expert divides $WS$ into two groups: (1) $R_{ws_i} \subseteq WS$: a subset including all the publicly removed world states, i.e., all the world states that the expert declared "cannot be the actual world state" and (2) $L_{ws_i} \subseteq WS$: a subset including all the publicly non-removed (left) world states, i.e., those that the expert has declared "might be the actual world state." Note that $R_{ws_i} + L_{ws_i} = WS$. The expert is assumed to be truth-telling, and hence we assume that $ws_i \in L_{ws_i}$.

Figure 1 depicts the flow of our model. First, nature sets the actual world state to be $ws_i$. Then, all players (expert, auctioneer, and bidders) receive information regarding the auction's characteristics $\left(n, WS, \forall ws_i \ p(ws_i), T, \forall t_j \ p(t_j), and \ V\right)$. Next, the expert becomes acquainted with the true world state (as noted, this might require some of their resources such as time or money). Based on the true world state, the expert divides the set of possible world states into $R_{ws_i}$ and $L_{ws_i}$. Finally, the expert uses signals to publicly disclose this division to both the auctioneer and bidders. Considering the set $L_{ws_i}$, the auctioneer then decides if they are interested in purchasing the information regarding the exact world state or not. If the auctioneer decides to purchase the information, $L_{ws_i}$ is updated to include only the actual world state and the bidders place their bids based on $ws_i$. If the information was not purchased, the bidders bid based on $L_{ws_i}$, which necessarily includes more than one member (being a self-interested player aiming to maximize the payment they receive for the information, we assume the minimal size of $L_{ws_i}$ to be two, as the strategy of disclosing the actual world state for free, i.e., the case where $|L_{ws_i}| = 1$, is dominated by all other possible strategies).

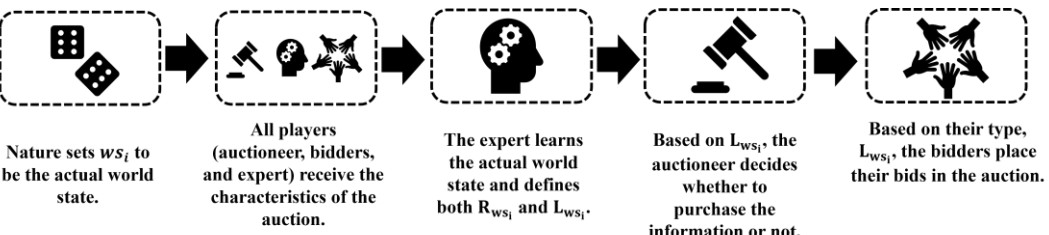

**Figure 1.** Flowchart of the model.

## 4. Preliminaries

Our study is motivated by the work of [7], which provides a full analytical analysis of the process for providing strategic information in the context of second-price auctions. Their work focuses on an environment in which all participants, i.e., the expert, the auctioneer, and the bidders, are completely rational agents. We, however, are interested in exploring a more realistic case in which some of the participants are human and therefore might not accommodate the rigid model of rationality. Specifically, we are interested in testing the effectiveness of strategic signaling in the case where human auctioneers are the target audience.

The following subsections provide a short summary of the equilibrium analysis described by [7], a discussion of humans' bounded rationality, and a description of all the adjustments required for the analysis to consider the case of a human auctioneer.

### 4.1. The Rational Agents Case

In games where all agents are completely rational agents, an equilibrium is calculated by finding a strategy profile in which all players are using their best response strategy. Being the ones who place a bid, the bidders have a direct effect on the auctioneer's strategy,

who in turn influences the expert's strategy. Thus, the analysis starts with finding the bidders' best response strategy.

In a second-price auction, the bidders' best response is necessarily to bid their true valuation [54], i.e., if the actual world state, $ws_i$, is known to the bidders, a bidder of type $j$ should bid $V[j][i]$. Otherwise, if the information is not purchased by the auctioneer, the bidders should update the probabilities assigned to each possible world state $ws_i \in WS$ based on the new information received, i.e., $L_{ws_i}$. Based on the updated (posterior) probabilities, the best response strategy is to bid their expected evaluation of the possible world states [22]. Therefore, bidder $j$'s optimal bidding strategy is influenced by their own type, $t_j$, and one of the two following components: (1) the actual world state, $ws_i$, disclosed to them by the auctioneer; or (2) the division into $R_{ws_i}$ and $L_{ws_i}$. Formally, it is captured by the function $B_{t_j} : WS \to \mathbb{R}$ as follows:

$$B_{t_j}(\alpha) = \begin{cases} V[j][i] & \text{if } \alpha = ws_i \\ \sum_{ws_k \in \alpha} p(ws_k \,|\, \alpha) \cdot V[j][k] & \text{if } \alpha = L_{ws_i} \end{cases} \tag{1}$$

where $p(ws_k|\alpha)$ is the conditional probability of the actual world state to be $ws_k$, given that the set of still possible world states equals to $\alpha$, specifically:

$$p(ws_k|\alpha) = \begin{cases} \frac{p(ws_k)}{\sum_{ws_z \in \alpha} p(ws_z)} & \text{if } ws_k \in \alpha \\ 0 & \text{otherwise} \end{cases} \tag{2}$$

Similar to the bidders, when deciding on their best response strategy, the auctioneer should consider the division into $R_{ws_i}$ and $L_{ws_i}$ made by the expert. However, the auctioneer also needs to consider both the payment requested by the expert and the best response strategy of the bidders. The function $EP_a : WS \to \mathbb{R}$ denotes the expected profit of the auctioneer from the auction (i.e., the second highest bid) when the expert divides $WS$ into $R_{ws_i}$ and $L_{ws_i}$, and the bidders use their best response strategy. Here again, if the information was purchased, the argument of the function is the true world state; if it was not, it is the set $L_{ws_i}$ (For the detailed calculation of $EP_a$ see [13]). Finally, the auctioneer's best response is to purchase the information whenever its value is greater than the asked cost. Formally, the information is purchased whenever:

$$\sum_{ws_k \in L_{wsi}} p(ws_k | L_{wsi}) \cdot EP_a(ws_k) - EP_a(L_{wsi}) \geq c \tag{3}$$

If the best response strategies of both the auctioneer and bidders are known, it is straightforward to calculate the best response strategy of the expert, who is aiming to maximizes their expected profit from the auction. The expert's strategy will therefore be to choose the set $L_{ws_i}$ which maximizes their expected profit, given by:

$$\underset{L_{wsi}}{\text{ArgMax}} \left( \sum_{ws_k \in L_{wsi}} p(ws_k | L_{wsi}) \cdot EP_a(ws_k) - EP_a(L_{wsi}) \right) \tag{4}$$

meaning that the expert reviews all of the possible legal (a legal division follows the rules mentioned in the model section, i.e., $R_{ws_i} + L_{ws_i} = WS$, $|L_{ws_i}| > 1$, and $ws_i \in L_{ws_i}$). divisions and for each division calculates the maximal price the auctioneer would be willing to pay for the information. The maximal price is calculated according to the difference between the auctioneer's revenue when having the information and their revenue when not having it. Finally, assuming $L'_{ws_i}$ is the subset that maximizes the above expression, the expert will set the price of the information as:

$$c = \sum_{ws_k \in L'_{ws_i}} p\left(ws_k \,\middle|\, L'_{ws_i}\right) \cdot EP_a(ws_k) - EP_a\left(L'_{ws_i}\right) \tag{5}$$

*4.2. Humans and Bounded Rationality*

Classic game theory in general, and auction theory specifically, assumes participants to be fully rational agents, i.e., they aim to act optimally given premises and information. As a result, most solutions to game theoretic problems focus on efficient policy solutions and the associated advantages of price incentives. In real life, however, it is well known that people have bounded rationality, e.g., a lack of knowledge or limited computational capacity. Moreover, people are easily affected by various external factors, such as cognitive bias, laziness, emotions, and social influences [53,55–59].

Behavioral economics [60–62] offers alternative, more realistic views on individual behavior. As a subfield of economics, behavioral economics focuses on the psychological, social, and emotional factors that influence people's decision-making processes. Behavioral economics is being applied in many fields, such as marketing [63,64], finance [65], and health [66]. Naturally, when it comes to setting policies in the real world, it is necessary to consider the broader picture, i.e., the fact that people do not always behave the way classic economics expects them to. Therefore, in this paper we consider a classic game-theory model and explore how human behavior, specifically of the auctioneer who faces the decision problem, affects the participants' behavior, strategies, and profit. For simplicity, we assume the bidders to be completely rational. Analyzing the auctioneer's behavior when facing bidders who are not completely rational is much more complicated, and requires generation of a (stochastic) behavior function to achieve a good-enough approximation of the bidders' bids.

*4.3. Changes and Necessary Adjustments*

In the general case, as presented in [7], the expert first commits to a signaling scheme, i.e., before getting acquainted with the actual world state, the expert publishes the set of signals they intend to use. Then, after learning about the actual world state, the expert transmits the suitable signal. Formally, a signaling scheme is composed by a set of possible signals to be used, $S^*$, and a signaling function, $S : WS \rightarrow S^*$, that matches signals for each possible world state. We emphasize that the expert is not obligated to use truthful signals. For example, if there are four possible world states, $\{ws_1, ws_2, ws_3, ws_4\}$, a possible signaling scheme can be $\{S^*, S\}$, where:

$$S^* = \{\{ws_3, \ ws_4\}, ws_2\}$$

$$S(ws_1) = \{ws_3, \ ws_4\}, \ S(ws_2) = ws_2, \ S(ws_3) = \{ws_3, \ ws_4\}, \ S(ws_4) = ws_2$$

meaning that the signals the expert can use are $\{ws_3, ws_4\}$ and $ws_2$. If the actual world state is $ws_1$ or $ws_3$, the signal $\{ws_3, ws_4\}$ will be used (the expert will say that the possible world states are $ws_3, ws_4$). This signal is truthful in the case where the actual world state is $ws_3$ and untruthful in the case where the actual world state is $ws_1$. Nonetheless, in the commitment case, the truthfulness of the signals is not important, as the expert's commitment to which signal will be used provides the auctioneer with more information than the actual content of the signal. In our paper, however, since we are dealing with people who have limited computational capacity, we attempt to eliminate any externalities that might be confusing. Therefore, we preset the scheme used by the expert to a more intuitive and natural one. The signaling scheme used in this study is the simplest version of the truthful elimination (TrE) variant mentioned by [67], in which the content of the signal stands for the still-possible world states (the tangible form of this signaling scheme is the division into $R_{ws_i}$ and $L_{ws_i}$). Examining additional, more complex, signaling schemes is definitely an interesting future research direction, as it may have unexpected influence on the auctioneer's behavior, which could lead to additional valuable insights.

## 5. Empirical Study

In this paper, we examine how the presence of a human auctioneer in the auction influences the existing theoretical results. For this purpose, we constructed, implemented,

and conducted a set of experiments that imitate a second-price auction setting, including several signaling strategies and several levels of awareness, as described in the following subsections.

### 5.1. Experimental Environment

The core of our experimental setup is a single-agent, multiround game called "The Mysterious Auction Game," which was inspired by the game "What's in the Box" presented in [53], which we adjusted to suit both the purpose and the domain of our research. In this game, the user is in the role of an auctioneer selling an item with an unknown value to a group of bidders in a second-price, sealed-bid auction. Note that although the game simulates an auction environment, which is a classic multiagent system, this is a single-agent game because the bidders are only reflected in their bids. As completely rational agents, their bids are equal to their true valuation of the auctioned item based on the possible world states as presented in $V$. In each round, a set of randomly generated game settings, including the number of bidders, the set of possible world states and their probabilities, the number of bidder types and their probabilities, and each type's valuation of each possible world state, are introduced to the user on-screen.

#### 5.1.1. Scoring System

As opposed to a real-life auctioneer, in our simulated auction environment, the auctioneer does not actually own the auctioned item, and therefore does not have an inner valuation of it. Specifically, their valuation of the item they offer for sale is zero, and consequently they will benefit from any positive payment offered them in exchange for the item. Since according to the definition of an auction, bidders' bids can be only positive, the auctioneer has no real incentive to pay the expert for the information, regardless of the bidders' bids. This lack of motivation for purchasing the information creates a degenerate version of a real-life auction. For our game to be a more accurate reflection of reality in which the users have real motivation to purchase the information, we added a risk factor in the form of a baseline. For each round, we calculate the value of the second highest bid to be placed in this round under uncertainty, denoted as $x_{baseline}$. In practice, the calculation of $x_{baseline}$, requires reviewing all bidder types and calculating the valuation expectancy for each one, assuming that all of the world states presented on-screen are possible. Having each type's bid, and considering both the different bidder types' probabilities and the number of bidders taking part in the auction, one can calculate the second-highest bid to be placed under uncertainty (see full details in [13]). Using $x_{baseline}$, we define the user's score for each round to be the difference between the actual second-highest bid placed in this round, denoted $x_{round}$, and $x_{baseline}$, minus the price of the information, if purchased. Note that because our model obligates an auctioneer who has purchased the information to disclose it to the bidders, once the information is purchased, the user assumes that the bidders also receive it, and hence their bids will be based on their valuation of the item considering the revealed world state. Therefore, as opposed to the value of $x_{baseline}$, which is not dependent on the identity of the actual world state, when the information has been purchased, the value of $x_{round}$ is directly affected by the identity of the actual world state and might be greater, smaller, or even equal to $x_{baseline}$; thus, the difference between $x_{round}$ and $x_{baseline}$ may be positive, negative, or zero respectively. Alternatively, if the user decided not to purchase the information, i.e., $x_{round} = x_{baseline}$, and no payment to the expert is required, their score for the round will equal zero. All details regarding both the nature of a second-price, sealed-bid auction and the scoring system were clearly and comprehensibly explained to the users at the beginning of the game.

#### 5.1.2. Game Flow

After being familiarized with the game settings and the scoring system, in each round, the user needs to decide if they are interested in purchasing the information revealing which of the world states presented on-screen is the actual one. The price for which the

information is offered for sale is also displayed on-screen, and can change from round to round. If one decides to purchase the information, the price requested is deducted from the user's accumulated score, and all of the untrue world states are crossed out, leaving the actual world state as the only one possible. Next, appropriate adjustments are made to the user's total accumulated game points, adding or subtracting game points depending on the difference between $x_{round}$ and $x_{baseline}$, and the user moves on to the next round. If the user decides not to purchase the information, however, no new information is provided to the user (or to the bidders) and their score stays unchanged. Finally, at the end of each round, the player receives a short summary noting the change in their accumulated game points. The goal of the user is to accumulate as many points as possible during the game.

### 5.2. Experimental Design

The Mysterious Auction Game was implemented using Django for the server side and JavaScript for the client side to allow the creation of a user-friendly graphic interface. To test the influence that signals have on human auctioneers, we constructed a set of 10,000 cases, each denoted by $e\langle Au, c, s, Aw \rangle$, where $Au$ is the auction setting, $c$ is the information cost as set by the expert, $s$ is the strategy the expert chose to use, and $Aw$ is the awareness level of the auctioneer. We built the set of experiments in stages, as follows.

1. Auction Setting ($Au$). Each auction setting includes the following parameters: (1) $n \in \{4, 5, 6, 7, 8\}$—the number of bidders who take part in the auction; (2) $|WS| \in \{3, 7\}$—the number of possible world states; (3) $|T| \in \{2, 5\}$—the number of possible bidder's types; and (4) $V$—the valuation matrix that holds each bidder's type valuation for each possible world state. All numerical values are expressed in terms of game points. Figure 2 depicts a screenshot of our experiment, demonstrating the settings that are introduced to the users. As part of our attempt to create an impartial decision-making process, we chose to eliminate any externalities that might distract the users and influence their decision-making process. Thus, we used uniform distribution for describing both each world state's probability of being the actual one and each bidder's type probability of being the actual bidder's type, i.e., $\forall ws_i \; p(ws_i) = \frac{1}{|WS|}$, and $\forall t_j \; p(t_j) = \frac{1}{|T|}$. With 5 options for the number of bidders, 2 options for the possible world states, and 2 options for the bidder types, we end up with $(5 \cdot 2 \cdot 2 =)20$ core cases, each of which includes a specific combination of parameters (1), (2), and (3). Parameter (4) will be discussed by the end of the next stage.

2. Information Cost ($c$). As explained in Section 4, a rational information buyer will be willing to purchase the information only if its cost is lower than its value, i.e., the value of the information is the maximal amount a rational information buyer will agree to pay in exchange for the information. Therefore, a strategic expert will set the information cost to be equal to the exact value of the information. This, however, might not be the case when facing a human information buyer. We denote the value of the information using $VoI(Au, i)$, where $Au$ is the auction setting and $i$ is the information to be disclosed by the expert (see Equation (3) for a complete calculation of $VoI(Au, i)$). To examine the effect that the extent of the difference between the value of information and its cost has on people and on their decision-making process, we set a range of possible information costs that can be used. For each core case defined, $Au$, given a specific piece of information the expert is interested in selling, $i$, we first calculate the numeric value of $VoI(Au, i)$. Then, we extend each core case into 4 cases differing from one another only in the cost of the information. In two of the cases, the cost is lower than the actual value of the information, i.e., the costs are equal to $0.5 \cdot VoI(Au, i)$ and $0.9 \cdot VoI(Au, i)$. In the other two cases, the cost is higher than the actual value of the information, i.e., the costs are equal to $1.1 \cdot VoI(Au, i)$ and $1.5 \cdot VoI(Au, i)$. This results in an overall of $(20 \cdot 4 =)80$ core cases to be examined for which 50% of the information should be purchased and 50% of it should not. Finally, we generate 25 random bidders' valuation matrices for each core case, creating $(80 \cdot 25 =)2000$

cases to be tested by human users. Each value of the bidders' valuation was drawn from the range of [0, 100]. All values chosen were integers.

3.  Expert's Strategy (*s*). We executed the basic version of The Mysterious Auction Game as presented in the previous subsection in three alternative, extended versions. The first version is the No-Signals (NS) version, in which no signals are used, and thus no free information is disclosed to the user before they are required to decide regarding acquisition of the information. The second version is the Random-Signals (RS) version in which signals are used, but the identity of the world states to be eliminated is decided randomly. The last version is the Greedy-Signals (GS) version in which signals are used and the identity of the world states to be eliminated is decided greedily, i.e., the expert chooses to eliminate the values that will lead to a maximal *VoI* for the set of remaining values.

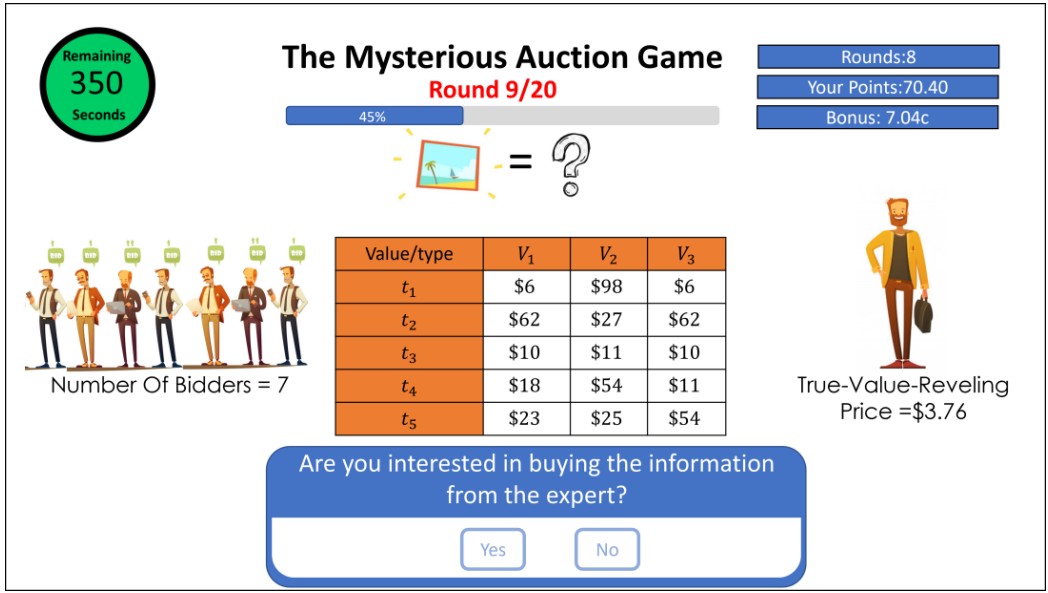

**Figure 2.** Screenshot of the Mysterious Auction Game. On the left side of the screen appears the number of bidders taking part in the auction. On the center of the screen appears a table including both the number of possible world states, the number of bidder types, and the matrix that contains the different types' valuation of the auctioned item for every possible world state. On the right side of the screen appears the price required for the information.

To allow users to capture the experience of being provided with signals in a realistic way, we first present all the possible world states on-screen. Then, after a few seconds, all the world states chosen to be eliminated (randomly or greedily) are crossed out by a red strikethrough so it is clear to the users that these are no longer valid options. Figure 3 depicts an example of three possible world states from which one world state (*v*3) is eliminated.

4   User Awareness (*Aw*). For the cases where signals are used, i.e., RS and GS, we considered two awareness alternatives:

*   Aware (A)—The user was informed that there is an additional player in the game, who gains from selling information. This was reflected both in the instructions provided to the users in the beginning of the game and through the user interface, so that for each round, the player's own accumulated score is displayed together with the expert's accumulated profit.
*   Unaware (U)—The user was told that the "system" is interested in assisting him by eliminating several untrue world states. No changes were made in either the instructions or the user interface.

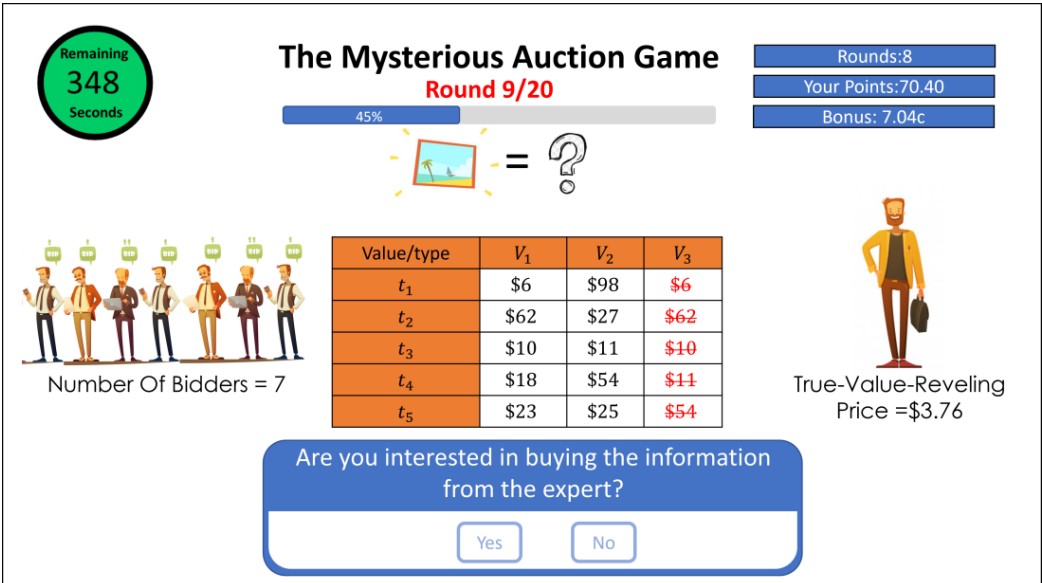

**Figure 3.** An example for how signals are being represented in The Mysterious Auction Game.

Overall, we have 5 different extended versions of our basic game: NS, RSA, RSU, GSA, and GSU. Since each version contains 2000 cases to be tested, we have a total of 10,000 cases. The full set of problem instances is available upon request from the corresponding author.

### 5.2.1. Recruitment of Participants

We recruited participants using the crowdsourcing marketplace Amazon Mechanical Turk (MTurk). MTurk is a popular method for data collection and is especially relevant for tasks that require human intelligence [68]. To avoid a carryover effect, each participant was allowed to participate only in one version of the game, i.e., we used a between-subjects design.

### 5.2.2. Participants' Compensation

We divided the compensation into two parts: (a) a "show-up fee"—all participants who completed the game, regardless of their achievements, received a payment of 40 cents; and (b) a "bonus"—to encourage thoughtful participation, a bonus payment was paid, as a direct outcome of the participant's performance. Users earned a bonus of 1 cent for every 10 game points accumulated.

### 5.2.3. Participants' Guidance

Before starting the game, the users were provided with a set of detailed instructions defining the rules of the game, their goal, and the compensation method to be used. To confirm their understanding of the game, the users had to complete a short quiz and were allowed to continue with the game only if they completed it successfully. After completing the quiz, users were asked to complete at least two trial rounds before starting the real game. Eventually, after completing both the quiz and the trial rounds, each user was asked to play a sequence of twenty rounds. The logic behind our choice to use a repeated game structure is similar to the one presented in [53], namely, when facing a set of different instances of the same decision problem, people tend to follow a decision rule based on expected monetary value (EMV). People's strategies do indeed asymptotically approach the EMV strategy as the number of instances increases [69–71]. Each round consisted of a problem instance randomly picked from the pool of 10,000 problem instances described above, with no repetition.

*5.3. Results*

In this section, we present a set of practical insights concerning the use of signals in auctions when the information buyer, i.e., the auctioneer, is human. Those insights will assist an expert acting in real-life environments to maximize their expected profit from the information they own. For our experiments, we recruited 500 participants, 100 for each experiment. Each participant played 20 rounds according to the above design. Participants' age ranged from 20 to 70 with an average age of 36.2, 58.2% were men and the rest women, 70% had a bachelor's degree, and 17% had a higher degree. All results presented in this section are statistically significant (*p*-value < 0.05) unless otherwise noted.

***Using greedy signals might lower people's willingness to purchase the information.*** Figure 4 depicts the information purchasing rate as a function of the number of possible world states, 3 or 7, and the number of possible bidder types, 2 or 5. In the NS experiment version when no free information was disclosed, the purchasing rate ranged from 66% to 68% throughout all of the different combinations of possible world states and bidder types (no statistical significance: *p*-value > 0.2). When there were 7 possible world states, we observed an interesting and surprising phenomenon: if greedy signals were used, i.e., if possible world states were eliminated such that the resulting *VoI* were the highest possible, the purchasing rate decreased substantially. In the GSU experiment version, the purchasing rate decreased to 60% and 63% for 2 and 5 bidder types, respectively. In the GSA experiment version, the purchasing rate decreased to 55% and 58% for 2 and 5 bidder types respectively. Those results are unexpected, as one would have imagined that the buyer would be more interested in purchasing the information when it was more valuable. When choosing to apply random signals, on the other hand, the results are not as conclusive, as in some cases the purchasing rate increased up to 72%, while in others it decreased to 62%.

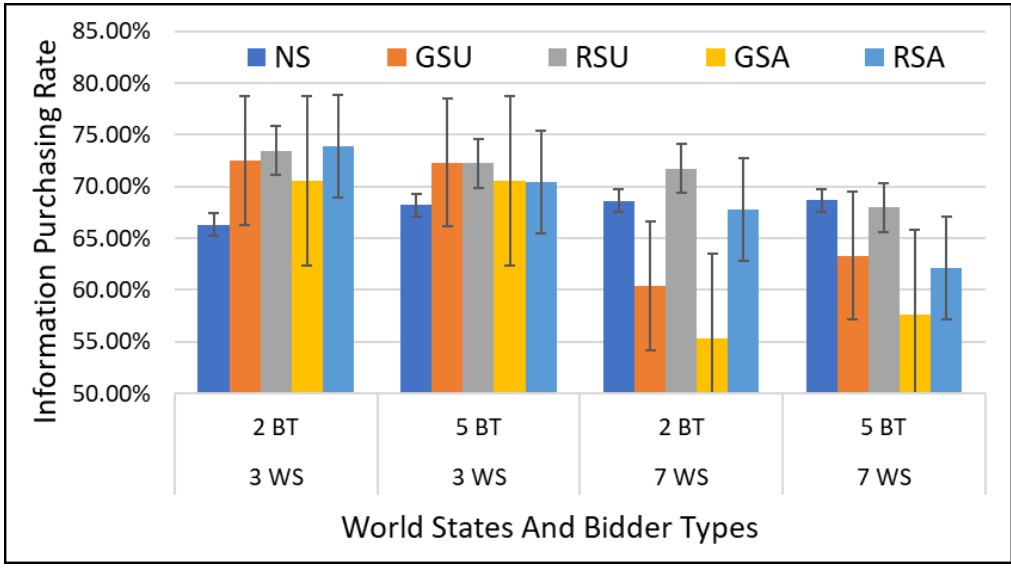

**Figure 4.** The information purchasing rate as a function of the number of possible world states and bidder types.

***The number of possible world states is a major factor in people's decision of whether to purchase the information or not, whereas the number of possible bidders' types is secondary for that decision.*** As can be seen in Figure 4, in the GSU experiment version, there was a difference between the purchasing rate in the case of 3 possible world states (72%, regardless of the number of bidder types) and 7 possible world states (60% and 63% for 2 and 5 possible bidder types, respectively, with no statistical significance between the two: *p*-value = 0.17). This was also the case in the GSA experiment version, in which the purchasing rate for 3 possible world states was 70% regardless of the number of the bidder types, and the purchasing rate for 7 possible world states was 55% and 58% for 2 and 5

bidder types, respectively (no statistical significance between the two: *p*-value = 0.23). If random signals were used when the auctioneer was unaware of the strategic nature of the expert, i.e., the RSU experiment version, no significant difference was found between the purchasing rate in most possibilities (71–73%). The only combination that had a statistically significant difference was when there were 7 world states and 5 bidder types, with purchasing rate of 68%. Finally, in the RSA experiment version, we observed a difference between 3 and 7 possible world states. Here also, in the case of 3 possible world states, the difference between the purchasing rate in the case of 2 bidder types (74%) and 5 bidder types (70%) was not statistically significant (*p*-value = 0.11). In the case of 7 possible world states, however, there was a statistically significant difference (*p*-value = 0.03) between 2 bidder types (68%) and 5 bidder types (62%).

*When facing scenarios with a small number of possible world states, people are not affected by the strategic nature of the expert.* As can be seen in Figure 4, in cases where there are only three possible world states, the differences between experiments GSA and GSU and between experiments RSA and RSU were relatively small (and not statistically significant: *p*-value > 0.12 and *p*-value > 0.51 for 2 and 5 bidder types, respectively). This was also the case when considering the expert's revenue. In Figure 5, which depicts the expert's average revenue as a function of the possible world states and bidder types, one can clearly observe that here too, the differences were relatively small and statistically insignificant (*p*-value > 0.26 and *p*-value > 0.37 for 2 and 5 bidder types, respectively) when considering 3 possible world states. The practical implication of this observation is that when there are only a few possible world states, both the purchasing rate and the revenue of the expert are identical, no matter whether the expert has disclosed themselves as a strategic player or not. Hence, a sudden discovery that they are facing a strategic expert will not change a human auctioneer's willingness to purchase the information or the price they are willing to pay. This, however, is not the case when facing a higher number of possible world states. In those cases, if they know that the expert is a strategic entity, the auctioneer will be willing to purchase the information in fewer cases and for lower prices. This is shown in both Figures 4 and 5 when considering the differences between experiments GSA and GSU and between experiments RSA and RSU. Both differences are shown in the figures and are statistically significant.

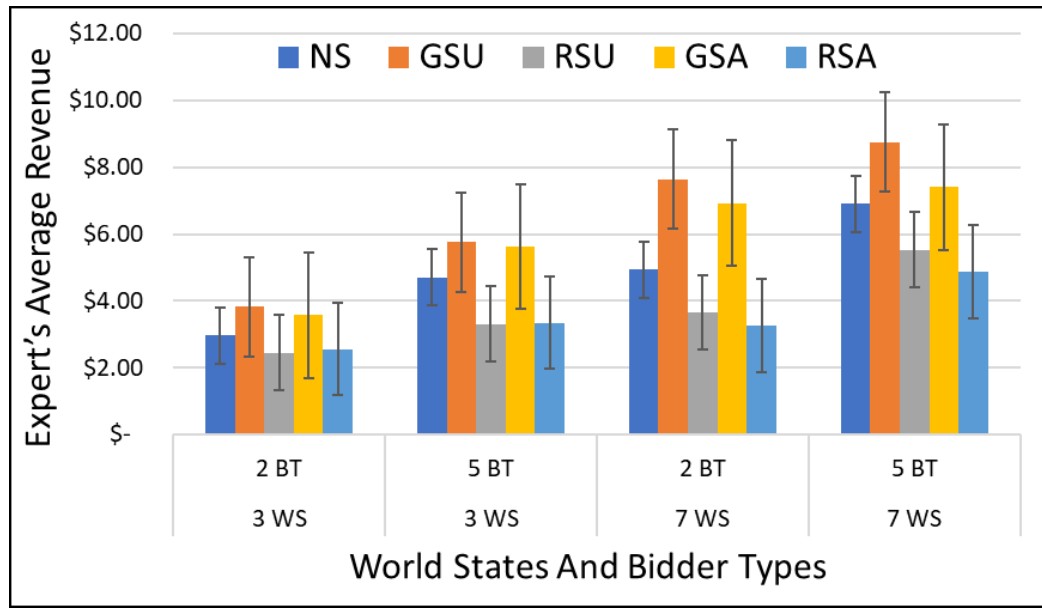

**Figure 5.** The expert's average revenue as a function of the number of possible world states and bidder types.

*Human auctioneers tend to purchase the information at a substantially higher rate compared to rational auctioneers.* Figure 6 depicts the information purchasing rate of a human auctioneer compared to the expected purchasing rate of a completely rational auctioneer for all different combinations of possible world states, bidder types, and experiment version. In the figure, it is evident that the purchasing rate in the case of a completely rational auctioneer stands around 50%, regardless of the number of possible world states, bidder types, and the experiment version performed (no statistical significance, *p*-value > 0.18). This observation does not come as a surprise, as each core case used in the experiments was extended into four cases, two of them with a price lower than the *VoI* (in which a rational agent will purchase the information) and the other two with a price higher than the *VoI* (in which a rational agent will not purchase the information). We note, however, that the percentages do not add up to exactly 50%, because in some cases (mostly in the case of 3 possible world states) the resulting *VoI* was zero, and therefore, no matter which of the coefficients was attached to the *VoI*, the resulting price was zero. In those cases where both the *VoI* and the cost of the information were equal to zero, a rational agent (i.e., the auctioneer) is indifferent to purchasing the information or not purchasing it. Thus, there are several scenarios in which those zero cases tipped the scale for the purchasing/not purchasing option, which led to an increase of one case over the other. On the other hand, from the figure, it shows that in all of the possible combinations, a human auctioneer tends to purchase the information for a substantially higher number of cases. Unlike a rational agent, a human auctioneer might purchase the information even in cases where its cost is higher than its value.

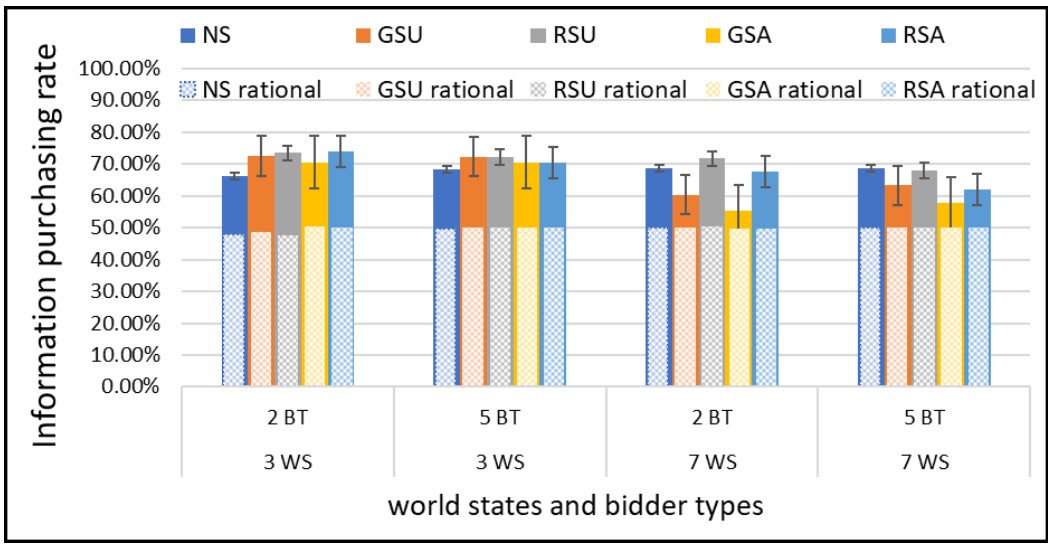

**Figure 6.** Information purchasing rate vs rational purchasing rate as a function of world states and bidder types.

*A low purchasing rate may result in a high monetary value for the expert.* Figure 7 depicts both the expert's average revenue and the information purchasing rate as a function of the *VoI* coefficient, i.e., the information cost for all experiment versions performed. As can be seen in the figure, as the information cost increases, the purchasing rate decreases. This does not come as a surprise since high purchasing cost tends to startle people presented with a purchasing transaction. However, what does seem a bit surprising is the fact that as the information cost increases (and thus the purchasing rate decreases), the expert's expected revenue also increases. This can be explained by noting that although the information is purchased in fewer cases, when it is purchased, it is bought at a relatively high cost. Therefore, the increase in the expert's expected revenue is a result of a delicate balance between the requested price and the number of cases in which the information is purchased. Obviously, this balance will not hold forever, and at some point (i.e., when the

price will become too high for the auctioneer, leading the auctioneer not to buy the information) it will break, and the expert's expected profit will drop. However, for the coefficients tested, high prices (which lead to low purchasing rate) result in high revenue for the expert. In addition, the inverse relationship between the purchasing rate and the information cost means that from the expert's point of view, if their main concern is exposure (e.g., they are interested in a high transaction volume to create a solid customer base), lowering the information cost is the preferable policy.

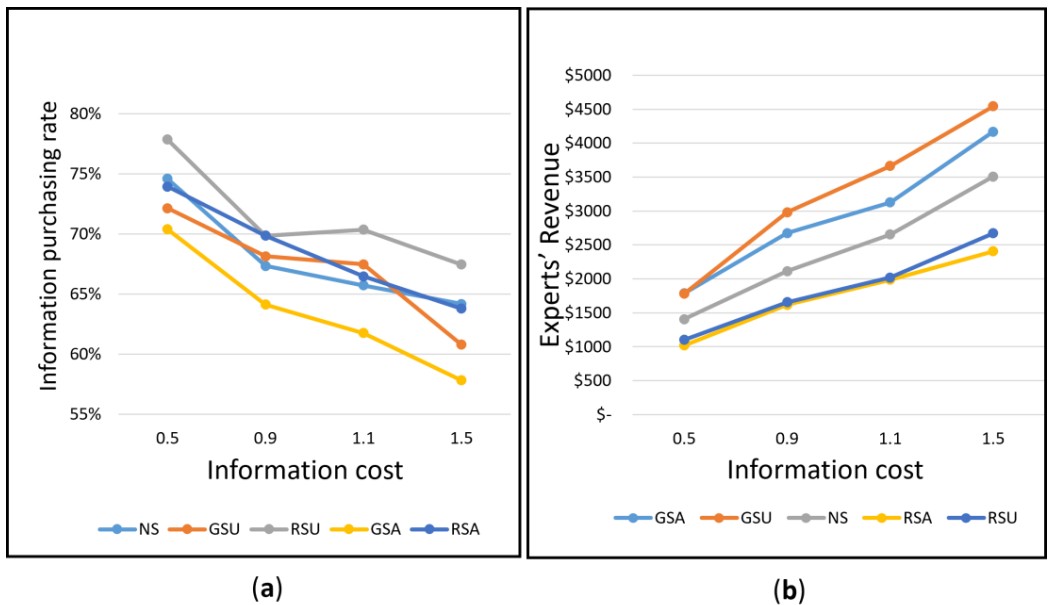

**Figure 7.** (**a**) Information purchasing rate as a function of the information cost. (**b**) Experts' revenue as a function of the information cost.

*Surprisingly, people are affected by the strategic nature of the expert when they use random signals, but not when they use greedy signals.* Figure 8 depicts a comparison between the information purchasing rates for the different *VoI* coefficients. From the figure, it is easy to see that if the expert uses greedy signals there is no difference between the GSU variant and the GSA variant, i.e., the auctioneers' behavior is identical, regardless of their awareness of the strategical nature of the expert. On the other hand, if the expert uses random signals, there is a difference between the RSU variant and the RSA variant in the 0.9–1.5 comparison, i.e., if the expert uses random signals, there are cases in which the auctioneer acts differently when they are aware of the strategic nature of the expert. This is very surprising, as one would have thought that if the strategic nature of the expert had an impact on the auctioneer, it would probably be when the expert is the one choosing the value of the signals, i.e., when they use greedy signals.

*People prefer to have information that is not useful, rather than risk the chance of not having valuable information.* As established in the analysis section, an auctioneer should purchase the information only if the *VoI* is higher than the information cost. Thus, an auctioneer's decision can fall into one of the following categories: (1) justified purchasing (JP)—when the *VoI* is higher than the information cost and the auctioneer's choice is to purchase the information; (2) unjustified purchasing (UJP)—when the *VoI* is lower than the information cost and the auctioneer's choice is to purchase the information; (3) justified avoidance (JA)—when the *VoI* is lower than the information cost and the auctioneer's choice is not to purchase the information; and (4) unjustified avoidance (UJA)—when the *VoI* is higher than the information cost and the auctioneer's choice is not to purchase the information. Figure 9 depicts the division of the auctioneer's decisions in the different experiments into these categories. From the figure, one can observe that the sum of the percentages of the times when the information was purchased, i.e., JP and UJP, ranged

between 63% to 70%, meaning that in a majority of the cases, the information was purchased. Moreover, examining the bad-decision categories (UJP and UJA) closely shows that the number of cases where the auctioneer should not have purchased the information but still chose to do so, i.e., UJP, was almost twice the number of cases where the auctioneer should have purchased the information, but chose not to, i.e., UJA. From these results, we can conclude that people are afraid of missing out, and thus if presented with the possibility of purchasing some allegedly useful information, they will many times choose to acquire it "just in case" without even pondering the value of the information offered. Finally, we noted that these results were consistent throughout all the different experiments, regardless of the type of signal used (if any) or the awareness of the auctioneer regarding the strategic nature of the expert.

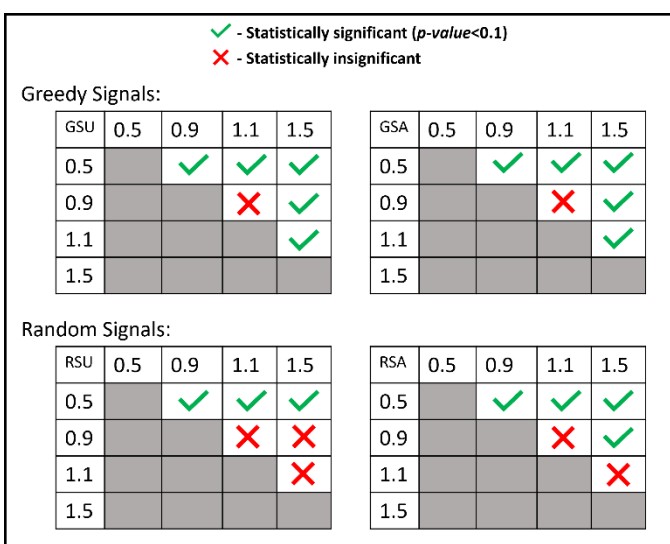

**Figure 8.** A comparison between the information purchasing rates for all different VoI coefficients.

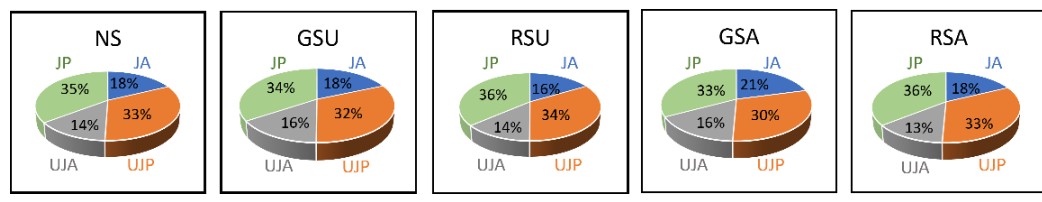

**Figure 9.** A numerical division of the users' decisions throughout the game into four categories: (1) justified purchasing (JP); (2) unjustified purchasing (UJP); (3) justified avoidance (JA); and (4) unjustified avoidance (UJA).

*There are cases in which human auctioneers' willingness to purchase the information decreases as the number of bidders in the auction increases*. Figure 10a depicts the information purchasing rate as a function of the number of bidders participating in the auction for all versions of the experiment. As can be seen in the figure, as the number of bidders increases, the purchasing rate decreases. This result is not always statistically significant (e.g., there is no statistical significance between the purchasing rate for the cases of 6, 7, and 8 bidders in the NS version), but there is a statistically significant difference between the cases of 4 and 8 bidders for all versions. This is depicted in Figure 10b. The described result is very surprising, since as the number of bidders increases, so does the probability that at least two bidders of the type who assign a relatively high value to the disclosed world state are participating in the auction, making the second-highest bid as high as possible. Therefore, the auctioneer's willingness to purchase the information should increase [13].

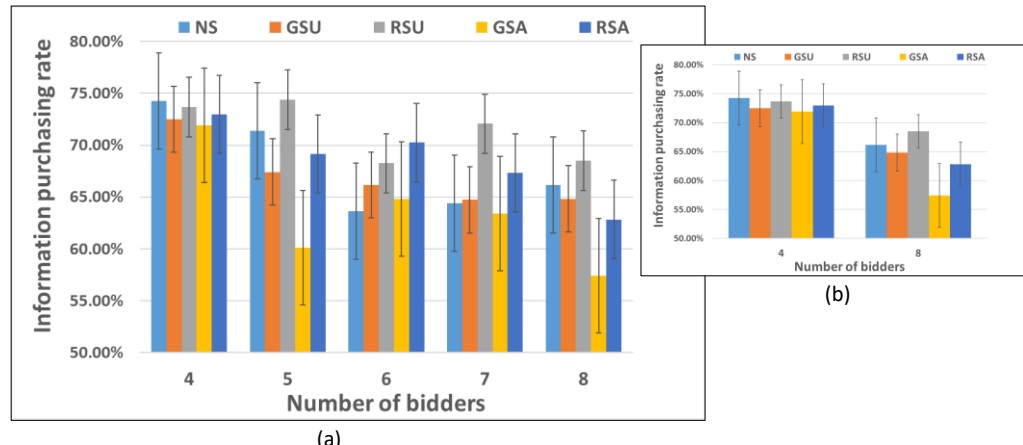

**Figure 10.** (**a**) Information purchasing rate as a function of the number of bidders participating in the auction. (**b**) A comparison between the information purchasing rate in an auction with 4 bidders and one with 8 bidders.

*In most cases, using random signals does not damage social welfare.* Figure 11 depicts the social welfare, i.e., the sum of the auctioneer's and the expert's revenue (since the focus of this paper is the behavior of a human auctioneer when a self-interested expert uses signals, we defined social welfare as the sum of the auctioneer's and expert's revenues; in future research, we might extend this definition to also include the bidders' revenue) as a function of the information cost for all five versions of the experiment. In the figure, it is evident that the resultant social welfare in experiment versions where the expert used greedy signals was relatively close to the social welfare resulting from using random signals (with no statistical significance). A statistically significant difference was observed only when the *VoI* coefficient equaled 1.5 and the user was unaware of the presence of a self-interested expert. In this case, the social welfare is higher if the expert uses greedy signals.

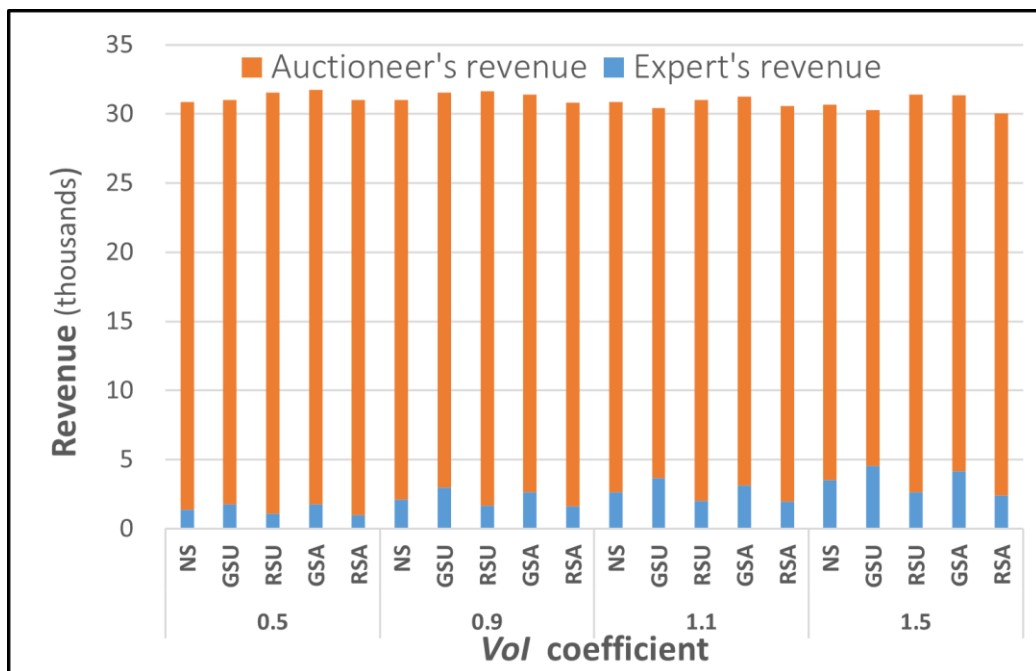

**Figure 11.** Social welfare as a function of the information cost.

## 6. Conclusions and Future Work

In this study, we investigated a new and more realistic perspective on the classic scenario in which an auctioneer in a second-price auction faces uncertainty regarding the value of the item they are offering for sale. We focused on a standard model in which a strategic external entity owns information that can disambiguate the auctioneer's uncertainty and aims to sell it for a fee. As opposed to previous studies that assumed the auctioneers to be completely rational agents, we relax this assumption and acknowledge the non-negligible number of cases in which humans fill the role of the auctioneer, e.g., e-commerce websites such as Amazon or eBay. This change is important, because people's rationality is bounded, and thus they might act differently from what one would expect. Consequently, a thorough investigation of how the use of signaling affects the different participants' behavior, strategies, and profit in such a case is required.

To obtain a better understanding of the influence signals have on human auctioneers, and gather practical insights that could serve as a toolbox for an expert acting in real-life auction environments, we conducted a comprehensive experimental evaluation of human behavior. For this purpose, we created 10,000 independent, simulated auctions differing from one another in these fundamental parameters: auction setting, information cost, expert strategy, and user awareness. Analyzing the results obtained, we found a wide range of significant, practical observations and insights that real-life experts could exploit to maximize their expected revenue from the information they own. While some of the results are similar to those presented in the rational auctioneer case, e.g., the information purchasing rate and the expert's revenue are inversely related, we also find some clear evidence that supports the existence of differences between humans and rational agents, e.g., human auctioneers, due to their tendency to prefer having useless information over the risk of losing valuable information, tend to purchase information in a relatively high number of cases compared to rational agents. Moreover, we showed that—as opposed to the classical auction theory—a human auctioneer does not always perceive having a higher number of bidders in the auction, and hence more competition, as an advantage. Those insights indicate that there is a gap between theory and human reality, and that understanding these biases can definitely help the expert in their decision-making process.

Throughout the paper, we assumed that the bidders were completely rational agents. This assumption, like the one made regarding the auctioneer, could also be relaxed. In future work, we plan to extend our model and explore cases in which not only the auctioneer but also the bidders have bounded rationality. Considering human bidders opens a window for examining the mutual influence between the auctioneer and bidders, enabling us the opportunity to explore how a strategic, self-interested expert can use it to their advantage.

**Author Contributions:** Conceptualization, resources, data curation, writing—original draft preparation, writing—review and editing, visualization, project administration, and funding acquisition, A.L. and S.A.; software, validation, and investigation, A.L.; methodology, formal analysis, and supervision, S.A. All authors have read and agreed to the published version of the manuscript.

**Funding:** This research received no external funding.

**Institutional Review Board Statement:** The study was conducted in accordance with the Declaration of Helsinki and approved by the Institutional Review Board of Ariel University (authorization AU-ENG-SA-20210703, date of first approval 3 July 2021).

**Informed Consent Statement:** Informed consent was obtained from all subjects involved in the study.

**Data Availability Statement:** The data that support the findings of this study are available on request from the corresponding author, S.A.

**Conflicts of Interest:** The authors declare no conflict of interest.

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
