# Peer review of "On the Reality of Signaling in Auctions"

_information, doi:10.3390/info13110549_

Round 1
Reviewer 1 Report
This is a nice-written paper that looks at the signalling and information revelation with imperfect rational agents. I do not have any major concerns or suggestions. My minor suggestion is to make better connection with existing Economics and Finance literature on signalling in auctions. For example, in the literature review section, when the authors say “Strategic signaling and optimal free disclosure of information in auctions have been studied widely in literature on both economics [8–10]…”, their reference are not on Economics research. Similar research on strategic signalling bidding was originally done by Fishman (1988), with more recent extension by Daniel and Hirshleifer (2018), Dodonova (2017) and Dodonova and Khoroshilov (2014). It would be nice if more economics papers related to signalling in auctions would be included in the literature review.
References:
“A Theory of Preemptive Takeover Bidding” Michael J. Fishman. RAND Journal of Economics, 1988, vol. 19, issue 1, 88-101
“A Theory of Costly Sequential Bidding”, Kent D Daniel, David Hirshleifer. Review of Finance, Volume 22, Issue 5, August 2018, Pages 1631–1665
“Preemptive bidding and Pareto efficiency in takeover auctions”. Dodonova, A. Economics Letters, 2017, 159, pp. 214–216
“Can preemptive bidding in takeover auctions be socially optimal? Yes it can” Dodonova, A., Khoroshilov, Y. North American Journal of Economics and Finance, 2014, 27, pp. 34–47
Author Response
Dear reviewer,
Thank you for the constructive review. Based on your comment we updated our references to include more economics papers related to signaling in auctions.
As required, changes in the manuscript body are highlighted using the MS Word Track Changes mechanism.
Reviewer 2 Report
The paper is very well written. I enjoyed reading and learning from the paper. I think the paper can be published as is.
Author Response
Dear reviewer,
Thank you for the positive review.